

# Variations in soil carbon, nitrogen, phosphorus and stoichiometry along forest succession in southern China

Shuai Ouyang [1,2], Wenhua Xiang [1,2,*], Mengmeng Gou[3], Pifeng Lei [1,2], Liang Chen [1,2], Xiangwen Deng [1,2], Zhonghui Zhao [1,2]

[1] Faculty of Life Science and Technology, Central South University of Forestry and Technology, Changsha, Hunan, 410004, China

[2] Huitong National Station for Scientific Observation and Research of Chinese Fir Plantation Ecosystems in Hunan Province, Huitong, Hunan 438107, China

[3] Chinese Research Academy of Environmental Science, Beijing, 100012, China

*Corresponding to*: Wenhua Xiang (xiangwh2005@163.com)

**Abstract.** Floristic composition changes during forest succession influence nutrient cycling. However, variation patterns of soil carbon (C), nitrogen (N), and phosphorous (P), and soil stoichiometry (C:N, C:P, and N:P ratios) along forest succession are controversial. In this study, soil samples were collected at depths of 0–10, 10–20, and 20–30 cm in three forests at early,

middle, and late succession stages. Concentrations of soil organic carbon (SOC), total nitrogen (TN), and total phosphorus (TP) were measured. As succession proceeds, concentrations and storage of SOC and TN exhibited a significant increasing tendency, while those of TP decreased significantly. The tendency was more pronounced ($P < 0.05$) in soil depths of 0–10 cm, implying that more SOC, TN, and TP were stored in the upper soil layer. The ratios of soil C:P and N:P increased far more than the C:N ratio with succession progress, indicating that forests in this region were primarily limited by P over time. The

C:N, C:P, and N:P ratios decreased from the upper to lower soil layers at almost all succession stages. A significantly positive correlation was found between the SOC, TN, and TP concentrations ($P < 0.05$), implying a relatively constrained C:N:P ratio in this subtropical forest soil. Stand density, litter, soil bulk density, pH, texture, and elevation significantly affected SOC, TN, and TP concentrations and their storage. The effects of succession stage, stand density, soil depth, and soil properties on soil nutrient balance should be considered in future studies.



## 1 Introduction

Soil organic carbon (SOC), total nitrogen (TN), and total phosphorus (TP) are very important indicators of soil fertility and productivity (Chapin, 2003; Andrews et al., 2004; Jiménez et al., 2011). For instance, SOC directly affects the production capacity of ecosystems, and acts as an indicator of ecosystem response to the environment (Chapin, 2003). Soil N and P are

the main nutrient elements for plant growth, which affect photosynthesis and other processes associated with primary production (Quilchano et al., 2008; Liu et al., 2013). Furthermore, soil N and P cycles are closely correlated with SOC cycles (Gao et al., 2014), and have the potential to mitigate the effects of global climate change (Lal, 2004; IPCC, 2007; Sardans et al., 2012). Soil stoichiometry (C:N:P ratios) is a good indicator of soil nutrient status, and is widely used as a powerful tool to advance our understanding of the interactions between the above-ground plant community and below-ground soil nutrient

characteristics in terrestrial ecosystems (Mooshammer et al., 2014; Zechmeister–Boltenstern et al., 2015). Therefore, a better understanding of the variations in SOC, N, P, and soil stoichiometry is important to assess soil nutrient status in forest ecosystems and sustainable development.

The concentrations and distribution of soil C, N, and P can differ between diverse plant communities due to different chemical traits and amount of litter and root returned to the soil (Zhao et al., 2010; Deng et al., 2013). Recent studies suggest

that forest succession, caused by both natural succession and management activity, could improve the soil C sequestration capacity (Chang et al., 2011; Deng et al., 2013). However, to date, general relationships between soil N and P variations and forest succession remains controversial. Studies have found that N increases (Deng et al., 2014) or decreases (Bond–Lamberty et al., 2006) with forest succession. Following succession, soil P has been found to decrease (Chen et al., 2000), increase (Wang et al., 2011), or remain unchanged (Wen et al., 2005).

It is anticipated that soil stoichiometry (C:N:P) will change with forest succession, due to changes in species composition, biomass production, and soil properties (Deng et al., 2013; Fanin et al., 2013). In recent years, more studies have been conducted to examine soil stoichiometry changes during secondary forest succession in the northern Loess Plateau, China (Jia et al., 2012; Deng et al., 2013; 2014). However, in subtropical southern China, while several studies have investigated changes in stoichiometry in the tissues of aboveground vegetation in secondary forests, soil stoichiometry dynamics with

succession have been poorly examined (Huang et al., 2013; Jiao et al., 2013; Jiang et al., 2017). Therefore, it is essential to

characterize the dynamics of soil C, N, and P and nutrient stoichiometry following succession of secondary forests in southern China.

In this study, we investigated the concentrations and storage of SOC, TN, and TP at depths of 0–10, 10–20, and 20–30 cm in subtropical forests in their early, middle, and late succession stages in southern China. Specifically, the objectives were: (1) to examine changes in SOC, TN, and TP concentration and storage patterns along secondary forest succession; (2) to reveal soil C, N, and P stoichiometry characteristics and nutrient limitation in secondary forests; and (3) to investigate the effects of stand characteristics, soil properties, and topography variables on SOC, TN, and TP concentrations and their storage in a subtropical forest.

## 2 Materials and methods

### 2.1 Study site description

The study site was located in the Dashanchong Forest Park (latitude 28° 23' 58″ N–28° 24' 58″ N, longitude 113° 17' 46″ E–113° 19' 08″ E), Changsha County, Hunan Province, China. The park is characterized by hilly topography, with an elevation ranging from 55 m to 260 m above mean sea level. The site resides within the central region of the humid mid–subtropical monsoon zone in China. Based on climate data obtained from 1954 to 2010, the mean annual air temperature is 17.3°C, and maximum and minimum temperatures are 39.8°C and −10.3°C, respectively. The mean annual precipitation is 1416.4 mm, and fluctuates between 936.4 mm and 1954.2 mm. The well–drained clay loam red soil developed from slate and shale parent rock, and is classified as a Alliti–Udic Ferrosol, corresponding to Acrisol in the World Reference Base for Soil Resource (IUSS Working Group WRB, 2006). The subtropical evergreen broadleaved forest is the climax vegetation of the region. Due to historical human disturbances, the forest park has a range of secondary forests at different succession stages (Ouyang et al., 2016).

### 2.2 Experiment design and soil sampling

Three secondary forests, representing different succession stages, were chosen in this study. (1) The early stage is a *Pinus massoniana – Lithocarpus glaber* forest (PM–LG) dominated by shade-intolerant coniferous species. (2) The middle stage is





a *Choerospondias axillaris* forest (CA) dominated by shade-intolerant deciduous broadleaf species. (3) The later stage is a *L. glaber – Cyclobalanopsis glauca* forest (LG–CG) dominated by shade-tolerant evergreen broadleaved species. A 1-ha permanent plot was established within each forest. Detailed stand characteristics of the three forests were described in our previous study (Ouyang et al., 2016).

5 Each 1-ha plot was divided into 100 10 × 10 m subplots. Within each subplot, soil samples were collected at depths of 0–10, 10–20, and 20–30 cm and taken to a laboratory to measure SOC, TN, and TP concentrations and pH values. Soil bulk density for each depth was measured in triplicate using a steel soil corer, 5.0 cm in diameter and 5.0 cm in length, at points adjacent to the soil sampling plots. Thus, one pit was dug to 30 cm depth at the center of a subplot and three soil samples were taken. The volume of each soil corer and soil dry mass, after oven-drying at 105°C for 48 h, were measured.

10 **2.3 Physical and chemical analysis**

Soil pH values were measured at a soil-to-water (deionized) ratio of 1:2.5 using an FE20 pH meter (Mettler Toledo, Shanghai, China). Soil texture, i.e., clay, silt, and sand contents, were determined using the pipette method (Gee and Bauder, 1986). Soil moisture contents were measured gravimetrically and expressed as percentages of soil water to dry soil weight. For SOC, TN, and TP, soils were manually sorted to visually remove stones, plant roots, and litter, and then sieved through a 15 2 mm mesh. SOC concentrations were determined using the $K_2Cr_2O_7/H_2SO_4$ oxidation method. TN concentrations were measured using a Semimicro-Kjeldahl method, and TP concentrations were measured by sodium hydroxide (NaOH) fusion and the Mo–Sb colorimetric method.

**2.4 Calculation of SOC, TN, and TP storage**

For each individual soil profile, the mass storage per area (Mg ha$^{-1}$) for SOC ($C_S$), TN ($N_S$), and TP ($P_S$) was calculated 20 using the following equations (Liu et al., 2011):

$$C_s = \sum_{i}^{n} [D_i \times SOC_i \times BD_i \times (1 - G_i)/100]/100 \qquad (1)$$





$$N_s = \sum_{i}^{n} [D_i \times TN_i \times BD_i \times (1 - G_i)/100]/100 \qquad (2)$$

$$P_s = \sum_{i}^{n} [D_i \times TP_i \times BD_i \times (1 - G_i)/100]/100 \qquad (3)$$

where $n$ is the number of soil layers; $i$ is the $i$th soil layer; $SOC_i$, $TN_i$, and $TP_i$ are the SOC, TN, and TP concentrations (g kg$^{-1}$)

in the $i$th soil layer, respectively; and $BD_i$, $G_i$, and $D_i$ are soil bulk density (g cm$^{-3}$), the proportion (%) of coarse (> 2 mm)

fragments, and the thickness (cm) in the $i$th layer, respectively. In this study, we calculated $C_S$, $N_S$, and $P_S$ for a depth of 30

cm using data from the three different soil layers (0–10, 10–20, and 20–30 cm).

## 2.5 Statistical analyses

Differences in SOC, TN, and TP concentrations, storage, and their ratios in the forests at a given soil depth were analyzed

using one-way analysis of variance (ANOVA), followed by Tukey's test. The effects of forest and soil depth on SOC, TN,

and TP concentrations, storage, and their ratios were tested using two–way analysis of variance (ANOVA). Multivariate

models were used to explain SOC, TN, and TP concentrations and their storage with three types of variables: (1) stand

factors, including Shannon index, tree species number, stand density, average stand DBH, evergreen proportion (E),

deciduous proportion (D), and litter biomass; (2) soil properties, including soil pH value, soil bulk density (BD), soil

moisture concentrations (MC), clay content, and silt content and (3) topography variables, including elevation and convexity.

All statistical analyses were performed using R 3.00 (R Development Core Team, 2012).

## 3 Results

### 3.1 Changes in SOC, TN, and TP concentrations with succession

Forest, soil depth and their interaction significantly affected SOC, TN, and TP concentrations (Table 2). SOC and TN

concentrations at all soil depths increased with forest succession (Fig. 1A and Fig. 1B). Mean SOC for 0–30 cm soil depth

increased from 12.61 g kg$^{-1}$ in the PM–LG forest to 18.61 g kg$^{-1}$ in the LG–CG forest, and mean TN increased from 1.22 g

kg$^{-1}$ in the PM–LG forest to 1.63 g kg$^{-1}$ in the LG–CG forest. In a given forest, SOC and TN concentrations decreased with

soil depth, with the highest value at a depth of 0–10 cm.

In contrast, TP concentrations at all soil depths decreased with forest succession (Fig. 1C). Mean TP concentrations for 0–30 cm depth decreased from 0.31 g kg$^{-1}$ in the PM–LG forest to 0.25 g kg$^{-1}$ in the LG–CG forest. No significant difference was found for TP concentrations between the CA and LG–CG forests. Along with succession, there was a pronounced decrease in TP concentrations at a soil depth of 0–10 cm (Fig. 1C). As soil depth increased, TP concentrations appeared to decrease.

### 3.2 Changes in SOC, TN, and TP storage with succession

Forest succession, soil depth, and their interaction significantly affected $C_S$, $N_S$ and $P_S$ (Table 1). $C_S$ and $N_S$ for three soil depths increased with forest succession (Fig. 2A and Fig. 2B). Mean $C_S$ for 0–30 cm depth increased from 45.95 Mg ha$^{-1}$ in the PM–LG forest to 62.46 Mg ha$^{-1}$ in the LG–CG forest, and mean $N_S$ increased from 4.49 Mg ha$^{-1}$ in the PM–LG forest to 5.44 Mg ha$^{-1}$ in the LG–CG forest. In a given forest, $C_S$ and $N_S$ decreased with soil depth, with the highest value at a soil depth of 0–10 cm.

Similar to TP concentrations, TP storage at all soil depths decreased with forest succession (Fig. 2C). Mean $P_S$ for 0–30 cm soil depth decreased from 1.11 Mg ha$^{-1}$ in the PM–LG forest to 0.87 Mg ha$^{-1}$ in the LG–CG forest, although no significant differences in $P_S$ values were found between the CA and LG–CG forests. The highest $P_S$ was in the topsoil (0–10 cm), and the lowest was in the 20–30 cm soil layer (Fig. 2C).

### 3.3 Variations in soil C:N, C:P, and N:P ratios with succession

Soil C:N, C:P, and N:P ratios were significantly affected by forest succession and soil depth. While interactions between forest succession and soil depth significantly affected C:P and N:P ratios, they did not significantly affect C:N ratios (Table 1).

Soil C:N ratios at all three depths increased with forest succession. Furthermore, C:N ratios in the LG–CG and CA forests were significantly higher than those in the PM–LG forest ($P < 0.05$) (Fig. 3A) for all soil depths. C:N ratios did not significantly differ between the LG–CG and CA forests at the three depths ($P > 0.05$). In all forests, the soil C:N ratios in the 0–10 cm soils were significantly higher than at depths of 10–20 and 20–30 cm (Fig. 3B).



Soil C:P and N:P ratios increased with forest succession and differed significantly between forests. Soil C:P and N:P ratios at depths of 0–10 and 10–20 cm in the LG–CG forest were significantly higher than those of the CA and PM–LG forests ($P <$ 0.05). However, at 20–30 cm depth, soil C:P and N:P ratios in the CA forest were significantly higher than those in the LG–CG and CA forests ($P < 0.05$) (Fig. 3B and 3C). In all forests, soil C:P ratios were higher at 0–10 cm depth than at lower soil layers. Soil C:P and N:P ratios did not significantly differ between 0–10 cm and 20–30 cm depths in the LG–CG and CA forests ($P > 0.05$). However, there were significant differences between the three soil depths in the LG–CG forest (Fig. 3B and 3C).

In the PM–LG and CA forests, the soil C:N:P ratios gradually decreased at soil depths of 0–10 cm and 10–20 cm, then increased at a soil depth of 20–30 cm. In the LG–CG forest, the C:N:P gradually decreased from the 0–10 to 20–30 cm depths (Table 2).

### 3.4 Factors affecting SOC, TN, and TP concentrations and their storage

Stem density, litter biomass, soil PH, bulk density, silt content, and elevation showed significant effects on both SOC concentrations and storage, while other variables were not significant (Table 4 and Table 5). The TN concentration and storage multivariate model indicated that stem density, litter biomass, soil PH, bulk density, silt content, and elevation had significant effects on both SOC concentrations and storage. Only the Shannon index significantly affected TN concentrations, while other predictors did not (Table 4 and Table 5). In comparison, the TP concentration and storage multivariate model indicated that the Shannon index, stem density, litter biomass, soil PH, bulk density, silt content, and clay content had a significant effect on both SOC concentrations and storage. Elevation and convexity significantly affected TP storage, while other variables did not (Table 4 and Table 5).

## 4 Discussion

### 4.1 Effects of forest types and soil depth on SOC, STN, and STP concentrations and storage

Plant species influence soil C, N, and P primarily through litter decomposition, root secretion, soil mineralization, and



contributions from soil fauna, insects, and microorganisms (Gao et al., 2014). Furthermore, plant species differ in their capacity to utilize and capture C, N, and P (Wang et al., 2013). In this study, forest, soil depth, and their interactions significantly affected the SOC, TN, and TP concentrations and storage (Table 2), indicating that forest types and soil depth were important factors affecting the soil nutrient distribution. A similar change trend between concentrations and storage was found for SOC, TN, and TP (Fig. 1 and Fig. 2).

Along the successional gradient, we found SOC and TN concentrations increased, but TP decreased. This result indicated that forest succession enhanced SOC and TN concentrations and storage (Fig. 1), in agreement with studies that found that vegetation succession increased SOC and TN concentrations and storage (Davis et al., 2007; Deng et al., 2013; 2014). This general trend is likely due to vegetation recovery facilitating SOC and TN accumulation from litter and root input to the soil (Tang et al., 2010; Deng et al., 2013). For instance, some research has indicated that late successional species stands produce more fine roots than those at earlier succession stages (Yang et al., 2010; Deng et al., 2014). Total standing fine root biomass and annual fine root production increased between an early successional stage PM–LG forest to a late successional stage LG–CG forest, and their turnover (dead roots) also increased with vegetation succession (Liu et al., 2014).

In contrast, TP concentrations declined along forest succession due to the demands of vegetation growth. P is derived primarily from rock weathering, leading to extremely P-deficient soils (Aerts and Chapin, 1999). P is also the primary limiting factor in red soil areas in southern China due to P absorption and sorption (Gao et al., 2014). Our results support the conclusion that P becomes increasingly limiting relative to N in forests over time (Walker et al., 1976; Wardle et al., 2004).

Soil depth significantly affected SOC and TN concentrations and storage in the present study (Table 1). SOC and TN had significantly different dynamics between topsoil and subsoil in all forests (Fig. 2). SOC and TN concentrations in the upper soil (0–10 cm) for all forests were significantly higher than that in lower soil. This result agrees with previous studies showing that surface soil more actively sequesters SOC and TN (Wu, 2004; Justine et al., 2017). Most natural biological processes in forest ecosystems occur on the soil surface, which results in the highest SOC and TN in the topsoil (Hooper et al., 2000; Tian et al., 2010; Justine et al., 2017). Unlike TN and SOC, there were no significant differences in TP between the upper soil (0–10 cm) and the subsoil (10–20 cm) layers in all forests, and especially for the LG–CG forest (Fig. 2). This observation suggests that TP remained relatively stable during the late succession stage. One possible explanation is that TP



is primarily derived from rock P, so P available to plants in the soil is extremely low in forested soils, and depends on

interactions with iron (Fe) and aluminum (Al) (Oelmann et al., 2011; Gao et al., 2014).

**4.2 Effects of forest types and soil depth on C:N, C:P, and N:P ratios**

In this study, soil C:N, C:P, and N:P ratios were significantly affected by forest succession, soil depths, and their interactions

(Table 1). Forest succession improved soil C:N, C:P, and N:P ratios The largest soil C:N, C:P, and N:P ratios of 11.81, 70.95,

and 6.25, respectively (Fig. 4A, 4B and 4C), were found in the 0–30 cm soil in the late successional stage LG–CG forest.

Previous studies showed that varying land use types have different soil C:N:P ratios (Li et al., 2012; Zhao et al., 2015). Li et

al. (2012) suggested that variations in soil CNP stoichiometry might result from different vegetation and land management

practices. Zhao et al. (2015) conjectured that only vegetation and plant communities affect soil nutrient stoichiometry. Fan et

al. (2015) demonstrated that soil depth and successional stage significantly influence soil CNP stoichiometry.

Generally, living organisms and soil have relatively stable C:N:P ratios (Zhao et al., 2015). Furthermore, key ecosystem

characteristics are determined by these ratios (Michaels et al., 2003). Globally, a well-balanced C:N:P ratio is 186:13:1 for

surface soil, i.e., 0–10 cm (Cleveland and Liptzin, 2007; Wang et al., 2014). A general C:N:P ratio is 134:9:1 for 0–10 cm

organic–rich soil and 60:5:1 for the entire soil depth (0–250 cm) in China (Tian et al., 2010). In this study, the mean C:N:P

ratio was 59.7:5.4:1 for the 0–30 cm depth, and the highest C:N:P (69.2:5.7:1) was found for the 0–10 cm depth (Table 3).

These ratios were far below the average C:N:P for China and globally. For 0–10 cm depth soils, the C:N, C:P, and N:P ratios

of 12.25, 69.25, and 5.69, respectively, were lower than the average Chinese values reported in Tian et al. (2010), which

were 14.4, 136, and 9.3, respectively. These differences might be due to soil samples containing humified litter in the

Cleveland and Liptzin (2007) and Tian et al. (2010) studies, resulting in relatively higher C:N, C:P and N:P ratios than in our

results.

C:N ratios in our study were nearly all > 10 in all forests and soil layers (Fig. 4A). A low C:N ratio (< 25) implies that soil

organic matter is accumulating slower than it is decomposing (Zhao et al., 2015) and that there is net mineralization of N in

the soil (Wei et al., 2009). A C:N ratio lower than 10 indicates that less organic matter is being merged into the soil (Saikh et

al., 1998; Yimer et al., 2007). Similar to Li et al. (2016), we found TP changed only slightly between forests succession types

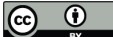

and soil depth; the soil C:P and N:P ratios were determined by the SOC and TN concentrations. The LG–CG (late succession) and CA (middle succession) forests had higher SOC and TN concentrations than the PM–LG (early succession) forest, resulting in higher C:P and N:P ratios, which is in agreement with the global soil nutrient ratios reported by Cleveland and Liptzin (2007). In addition, the topsoil layer 0–10 cm had greater soil C:N, C:P, and N:P ratios than the subsoil or deeper soil

because the litter layer released more nutrients into the topsoil (Li et al., 2016). The soil C:P ratios ranged from 42.91 to 70.95 for the 0–30 cm soil depth in this study, which implied a net mineralization of soil nutrients (< 200) (Pau, 2007). Although the total soil N:P ratios did not exceed the Redfield ratio of 16, the threshold postulated for P limitation in plankton (Jia, 2013), all N:P ratios in fresh leaves in all forests were greater than 16 and P concentrations were < 1.0 mg g$^{-1}$, indicating P limitation (Zeng et al., 2017).

Our study supported the hypothesis that C:N:P Redfield-like ratios are common in forest soil (Cleveland and Liptzin, 2007). The SOC, TN, and TP concentrations were significantly correlated (Table 3), and the relatively high correlation coefficient (0.83) for C and N concentrations indicated that the C:N ratio was highly constrained. Relatively constrained C:P and N:P ratios were also observed based on the correlation coefficients of 0.26 for the C and P concentrations, 0.30 for the N and P concentrations. Together, these implied a relatively constrained C:N:P ratio in subtropical forest soil, which is similar to that

reported in Cleveland and Liptzin (2007) and Tian et al (2010).

### 4.3 Factors affecting SOC, TN, and TP concentrations and their storage

SOC and soil nutrient concentrations and their storage usually vary depending on stand characteristics, soil properties, e.g., bulk density, texture, moisture, and pH value, and topography (Xia et al., 2016; Jiang et al., 2017). We found stand density, litter biomass, soil bulk density, pH, texture, and altitude did significantly affect SOC, TN, and TP concentrations and their

storage (Table 4 and Tables 5). In our previous study, we found stand density has a significant effect on stand biomass in this study area (Ouyang et al., 2016), which could create variations in SOC, TN, and TP concentrations and their storage. In natural succession, more plant litter decomposes and is then transformed into soil organic matter fallowing vegetation recovery and additional accumulation of plant litters (Zhao et al., 2010). Soil texture, pH, and soil moisture also contribute to variations in SOC, N, and P concentrations and their storage (Liu et al. 2015; Jiang et al., 2017). Soil bulk density

measurements were significantly related to SOC, TN, and TP concentrations and their storage. This correlation is consistent

with those previous reported in Liao et al. (2012) and Deng et al. (2013). The relatively higher SOC and TN concentrations

in clayey soils may be due to more decomposed organic matter and stabilization of clay particles in the soils (Leifeld et al.,

2005; Puget and Lal, 2005). Liu et al. (2015) found that soil texture was a key factor affecting variations in SOC and soil

nutrients under Karst topography. Yuan et al. (2013) found that soil moisture and pH value could explain 51% of observed

SOC variability in a temperate forest. Soil pH can affect the biogeochemical cycles of SOC and other nutrients by changing

microbial community activity, leading to variations in C, N, and P concentrations in forest soil (Cleveland and Liptzin, 2007;

Deng et al., 2013). However, no significant relationships were observed between soil moisture and SOC and TN

concentrations in our study. This agrees with the conclusions of Deng et al. (2013), who also found no significant

relationships between soil moisture and SOC during vegetation succession for humid climate conditions with annual rainfall

above 600 m in China. It is well known that topography, e.g., elevation and convexity, can affect local microclimates, litter

decomposition, and leaching of soil surface nutrients, resulting in large variations in SOC and soil nutrients (Xia et al., 2016;

Jiang et al., 2017). This relationship is especially clear in our study site under mountainous conditions, where topographic

gradients are variable (Yuan et al. 2013; Xia et al., 2016).

**5 Conclusions**

Our results support the hypothesis that forest succession significantly affects SOC, TN, and TP concentrations, storage, and

stoichiometry. Forest succession improved SOC and TN concentrations and storage, but decreased TP concentrations and

storage. In addition to forest succession, stand density, litter, soil properties, e.g., soil bulk density, pH, and texture, and

typography could significant affect SOC, TN, and TP concentrations and storage. Moreover, soil C:P and N:P ratios

significantly increased far more than N:P ratios with succession, indicating an increasing P limitation for plant growth as

succession proceeded in this region. These results enrich our current knowledge of C, N, and P soil stoichiometry in global

forest ecosystems, and provide useful information for sustainable forest management in southern China.

**Acknowledgements**

This work was supported by the National Key Research and Development Program of China (2016YFD0600202), National Natural Science Foundation of China (31570447 and 31700636), China Postdoctoral Science Foundation (2017M612605), and Huitong Forest Ecological Station Program funded by the State Forestry Administration of China. We would also like to

thank the administrative staff from the Dashanchong Forest Farm for their local support of this study.

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





**Table 1** The effects of forest, soil depth, and stand × depth interaction on SOC, TN, and TP concentrations and storage, and C:N, C:P, and N:P ratios. *P* values from two–way ANOVA tests are presented. Bold font indicates significant differences at *P* < 0.05.

| Factor | SOC | TN | TP | $C_S$ | $N_S$ | $P_S$ | C:N | C:P | N:P |
|---|---|---|---|---|---|---|---|---|---|
| Stand | **0.0000** | **0.0000** | **0.0000** | **0.0000** | **0.0000** | **0.0000** | **0.0000** | **0.0000** | **0.0000** |
| Depth | **0.0000** | **0.0000** | **0.0000** | **0.0000** | **0.0000** | **0.0000** | **0.0000** | **0.0000** | **0.0000** |
| Stand × depth | **0.0026** | **0.0421** | **0.0324** | **0.0000** | **0.0000** | **0.0000** | 0.0614 | **0.0000** | **0.0000** |

SOC, TN, TP, Cs, Ns, Ps, C:N, C:P, N:P, BD, and MC denote soil organic carbon, soil total nitrogen, soil total phosphorus, storage of soil organic carbon, storage of soil total nitrogen, storage of soil total phosphorus, C:N ratio, C:P ratio, N:P ratio, soil bulk density, and soil moisture concentrations, respectively.



**Table 2** Stoichiometry of soil C:N:P in three forest types in southern China.

| Soil depth (cm) | C:N:P of different forest types | | | |
|---|---|---|---|---|
| | Early (PM–LG) | Middle (CA) | Late (LG–CG) | Average |
| 0–10 | 47.5:4.3:1 | 70.7:5.7:1 | 89.3:7.0:1 | 69.2:5.7:1 |
| 10–20 | 36.7:3.8:1 | 59.4:5.4:1 | 65.7:6.3:1 | 53.9:5.2:1 |
| 20–30 | 44.4:4.5:1 | 65.5:6.1:1 | 57.8:5.4:1 | 55.9:5.3:1 |
| Mean | 42.9:4.2:1 | 65.2:5.7:1 | 70.9:6.2:1 | 59.7:5.4:1 |

PM–LG, *Pinus massoniana – Lithocarpus glaber* forest; CA, *Choerospondias axillaris* forest; LG–CG, *L. glaber – Cyclobalanopsis glauca* forest.



**Table 3** Correlation coefficients of soil C, N, and P concentrations, storage, and stoichiometry in subtropical forests in southern China.

| Properties | SOC | TN | TP | $C_S$ | $N_S$ | $P_S$ | C:N | C:P | N:P |
|---|---|---|---|---|---|---|---|---|---|
| TN | 0.83** | 1.00** | | | | | | | |
| TP | 0.26** | 0.30** | 1.00** | | | | | | |
| $C_S$ | 0.91** | 0.74** | 0.25** | 1.00** | | | | | |
| $N_S$ | 0.70** | 0.87** | 0.28** | 0.82** | 1.00** | | | | |
| $P_S$ | 0.06 | 0.08 | 0.81** | 0.24** | 0.31** | 1.00** | | | |
| C:N | 0.57** | 0.06 | 0.03 | 0.53** | −0.01 | −0.04 | 1.00** | | |
| C:P | 0.81** | 0.62** | −0.32** | 0.72** | 0.49** | −0.42** | 0.55** | 1.00** | |
| N:P | 0.58** | 0.71** | −0.42** | 0.50** | 0.60** | −0.50** | 0.02 | 0.83** | 1.00** |

SOC, TN, TP, Cs, Ns, Ps, C:N, C:P, N:P, BD, and MC denote soil organic carbon, soil total nitrogen, soil total phosphorus, storage of soil organic carbon, storage of soil total nitrogen, storage of soil total phosphorus, C:N ratio, C:P ratio, N:P ratio, soil bulk density, and soil moisture concentrations, respectively. * Significant difference at $P < 0.05$; ** Significant difference at $P < 0.01$.



**Table 4** Multiple linear regression of SOC, TN, and TP concentrations with stand characteristics, topography variables, and soil properties in 0-30 cm soil layers in subtropical forests in southern China. Stand characteristics are Shannon index, tree species number, stand density, average stand DBH, evergreen proportion (E), deciduous proportion (D), and litter biomass. Soil properties are soil pH value, soil bulk density (BD), soil moisture concentrations (MC), clay content, and silt content. The topography variables are elevation and convexity. Explanatory terms are listed in order of entering the models, and type I sums of squares are reported. DF, degree of freedom; MS, mean squared. Values in bold font indicate significant effect at $P$ < 0.05.

| Variables | SOC | | | | TN | | | | TP | | | |
|---|---|---|---|---|---|---|---|---|---|---|---|---|
| | DF | MS | F | P | DF | MS | F | P | DF | MS | F | P |
| Shannon index | 1 | 121.90 | 3.27 | 0.072 | 1 | 1.20 | 6.38 | **0.012** | 1 | 0.03 | 4.91 | **0.028** |
| Richness | 1 | 34.70 | 0.93 | 0.335 | 1 | 0.13 | 0.69 | 0.408 | 1 | 0.00 | 0.11 | 0.746 |
| Density | 1 | 508.40 | 13.64 | **<0.001** | 1 | 1.79 | 9.49 | **0.002** | 1 | 0.08 | 13.65 | **<0.001** |
| DBH | 1 | 90.60 | 2.43 | 0.120 | 1 | 0.11 | 0.59 | 0.444 | 1 | 0.02 | 3.82 | 0.052 |
| E | 1 | 19.50 | 0.52 | 0.470 | 1 | 0.01 | 0.06 | 0.811 | 1 | 0.00 | 0.00 | 0.995 |
| D | 1 | 64.50 | 1.73 | 0.189 | 1 | 0.20 | 1.05 | 0.307 | 1 | 0.01 | 1.07 | 0.302 |
| Litter biomass | 1 | 157.50 | 4.23 | **0.041** | 1 | 0.73 | 3.89 | **0.049** | 1 | 0.04 | 7.00 | **0.009** |
| PH | 1 | 791.80 | 21.25 | **<0.001** | 1 | 5.28 | 28.10 | **<0.001** | 1 | 0.03 | 4.84 | **0.029** |
| BD | 1 | 1171.70 | 31.44 | **<0.001** | 1 | 4.96 | 26.37 | **<0.001** | 1 | 0.05 | 8.31 | **0.004** |
| MC | 1 | 6.10 | 0.16 | 0.686 | 1 | 0.33 | 1.75 | 0.187 | 1 | 0.00 | 0.08 | 0.773 |
| Clay | 1 | 321.00 | 8.62 | **0.004** | 1 | 1.10 | 5.85 | **0.016** | 1 | 0.06 | 10.58 | **<0.001** |
| Silt | 1 | 0.80 | 0.02 | 0.887 | 1 | 0.55 | 2.94 | 0.087 | 1 | 0.10 | 18.57 | **<0.001** |
| Elevation | 1 | 204.30 | 5.48 | **0.020** | 1 | 5.07 | 26.96 | **<0.001** | 1 | 0.01 | 1.58 | 0.210 |
| Convexity | 1 | 20.20 | 0.54 | 0.462 | 1 | 0.02 | 0.10 | 0.751 | 1 | 0.00 | 0.01 | 0.920 |
| Residuals | 260 | 37.30 | | | 260 | 0.19 | | | 260 | 0.01 | | |



**Table 5** Multiple linear regression of soil organic carbon storage ($C_S$), nitrogen storage ($N_S$), and phosphorus storage ($P_S$) with stand characteristics, topography variables, and soil properties in 0-30 cm soil layers in subtropical forests in southern China. Stand characteristics are Shannon index, tree species number, stand density, average stand DBH, evergreen proportion (E), deciduous proportion (D), and litter biomass. Soil properties are soil pH value, soil bulk density (BD), soil moisture concentrations (MC), clay content, and silt content. The topography variables are elevation and convexity. Explanatory terms are listed in order of entering the models, and type I sums of squares are reported. DF, degree of freedom; MS, mean squared. Values in bold font indicate significant effect at $P < 0.05$.

| Variables | $C_S$ | | | | $N_S$ | | | | $P_S$ | | | |
|---|---|---|---|---|---|---|---|---|---|---|---|---|
| | DF | MS | F | P | DF | MS | F | P | DF | MS | F | P |
| Shannon index | 1 | 544.00 | 2.08 | 0.150 | 1 | 2.54 | 1.67 | 0.197 | 1 | 0.30 | 8.17 | **0.005** |
| Richness | 1 | 734.00 | 2.81 | 0.095 | 1 | 5.37 | 3.54 | 0.061 | 1 | 0.00 | 0.02 | 0.883 |
| Density | 1 | 6357.00 | 24.34 | **<0.001** | 1 | 35.45 | 23.36 | **<0.001** | 1 | 1.63 | 43.70 | **<0.001** |
| DBH | 1 | 321.00 | 1.23 | 0.269 | 1 | 1.09 | 0.72 | 0.397 | 1 | 0.00 | 0.02 | 0.877 |
| E | 1 | 189.00 | 0.72 | 0.396 | 1 | 1.52 | 1.00 | 0.318 | 1 | 0.00 | 0.00 | 0.995 |
| D | 1 | 37.00 | 0.14 | 0.707 | 1 | 0.06 | 0.04 | 0.840 | 1 | 0.01 | 0.22 | 0.638 |
| Litter biomass | 1 | 897.00 | 3.44 | **0.065** | 1 | 8.45 | 5.57 | **0.019** | 1 | 0.22 | 5.90 | **0.016** |
| PH | 1 | 2672.00 | 10.23 | **0.002** | 1 | 7.10 | 4.68 | **0.031** | 1 | 0.76 | 20.31 | **<0.001** |
| BD | 1 | 1835.00 | 7.03 | **0.009** | 1 | 10.72 | 7.04 | **0.001** | 1 | 3.44 | 92.62 | **<0.001** |
| MC | 1 | 586.00 | 2.24 | 0.135 | 1 | 0.52 | 0.35 | 0.557 | 1 | 0.01 | 0.18 | 0.670 |
| Clay | 1 | 3787.00 | 14.50 | **<0.001** | 1 | 17.95 | 11.83 | **<0.001** | 1 | 0.24 | 6.35 | **0.012** |
| Silt | 1 | 258.00 | 0.99 | 0.321 | 1 | 3.51 | 2.32 | 0.129 | 1 | 0.18 | 4.84 | **0.029** |
| Elevation | 1 | 4396.00 | 16.83 | **<0.001** | 1 | 8.88 | 5.85 | **0.021** | 1 | 0.20 | 5.34 | **0.022** |
| Convexity | 1 | 22.00 | 0.09 | 0.769 | 1 | 0.09 | 0.06 | 0.812 | 1 | 0.27 | 7.29 | **0.007** |
| Residuals | 260 | 261.00 | | | 260 | 1.52 | | | 260 | 0.04 | | |


**Figure Captions**

**Fig. 1** Mean values of SOC (A), TN (B), and TP (C) concentrations for soils in different forests, bars indicate the standard error. Different lowercase letters indicate significant differences ($P < 0.05$) among forests at a given soil depth. PM–LG, *Pinus massoniana – Lithocarpus glaber* forest; CA, *Choerospondias axillaris* forest; LG–CG, *L. glaber – Cyclobalanopsis*

*glauca* forest.

**Fig. 2** Mean values of SOC (A), TN (B), and TP (C) storage for soils in different forests, bars indicate the standard error. Different lowercase letters indicate significant differences ($P < 0.05$) among forests at a given soil depth. PM–LG, *Pinus massoniana – Lithocarpus glaber* forest; CA, *Choerospondias axillaris* forest; LG–CG, *L. glaber – Cyclobalanopsis glauca*

forest.

**Fig. 3** Mean values of C:N (A), C:P (B), and N:P (C) ratio for soils in different forests, bars indicate the standard error. Different lowercase letters indicate significant differences ($P < 0.05$) among forests at a given soil depth. PM–LG, *Pinus massoniana – Lithocarpus glaber* forest; CA, *Choerospondias axillaris* forest; LG–CG, *L. glaber – Cyclobalanopsis glauca*

forest.





**Figure 1**

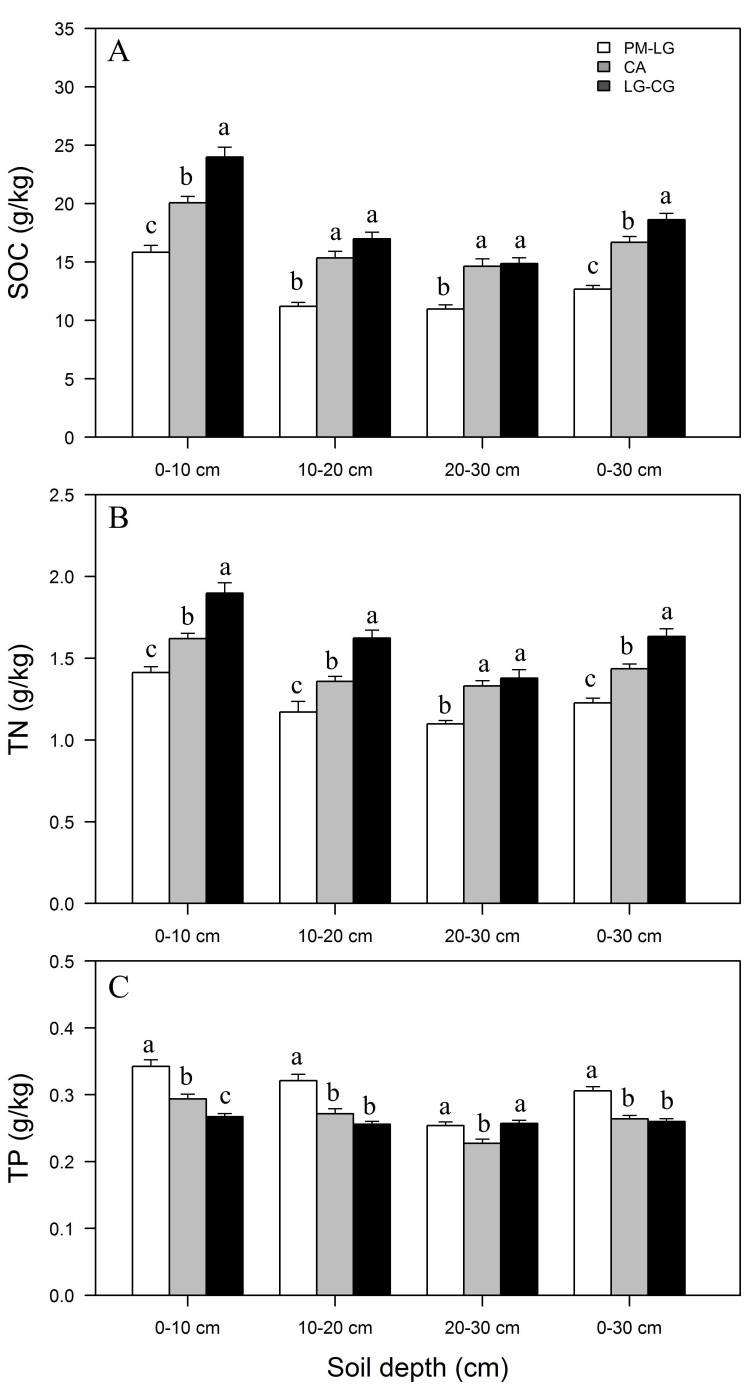



**Figure 2**

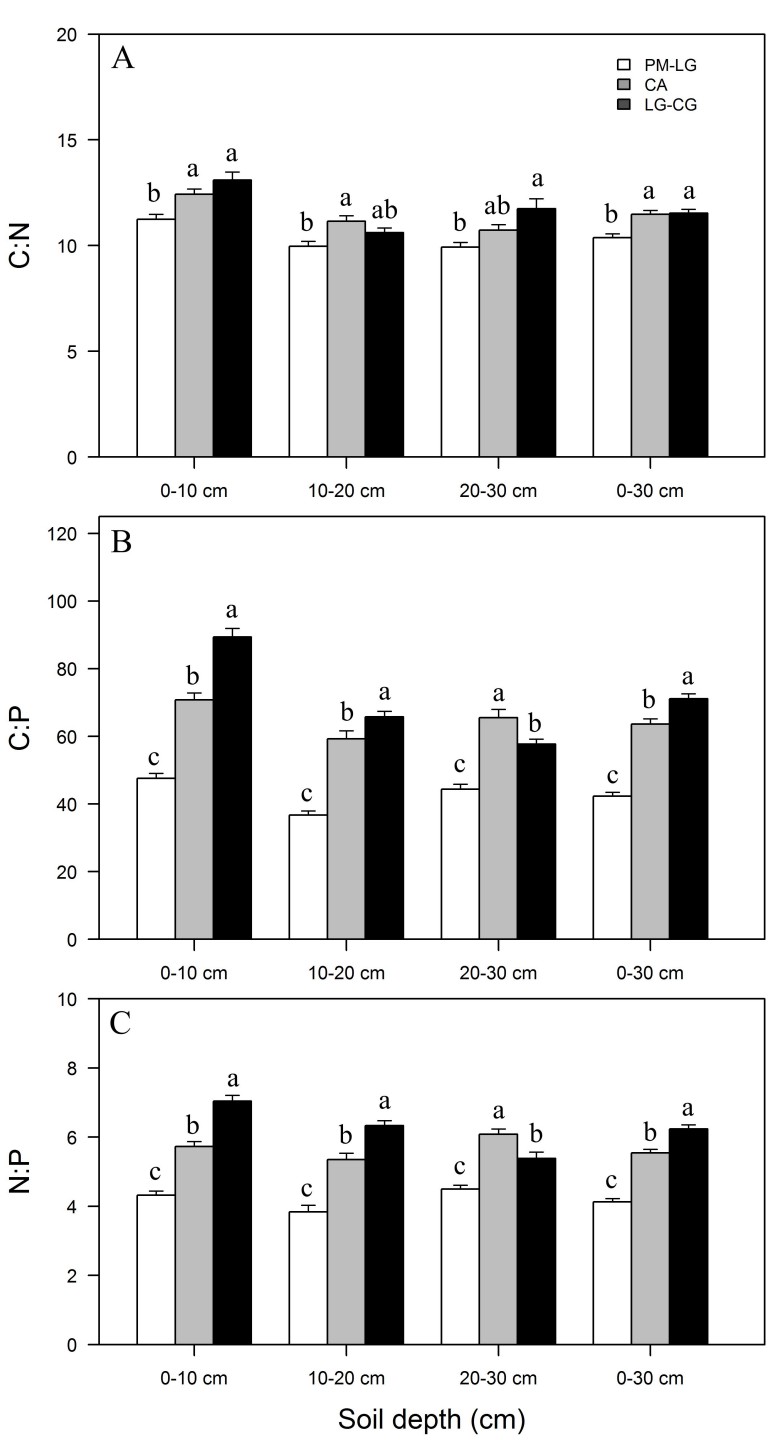



**Figure 3**

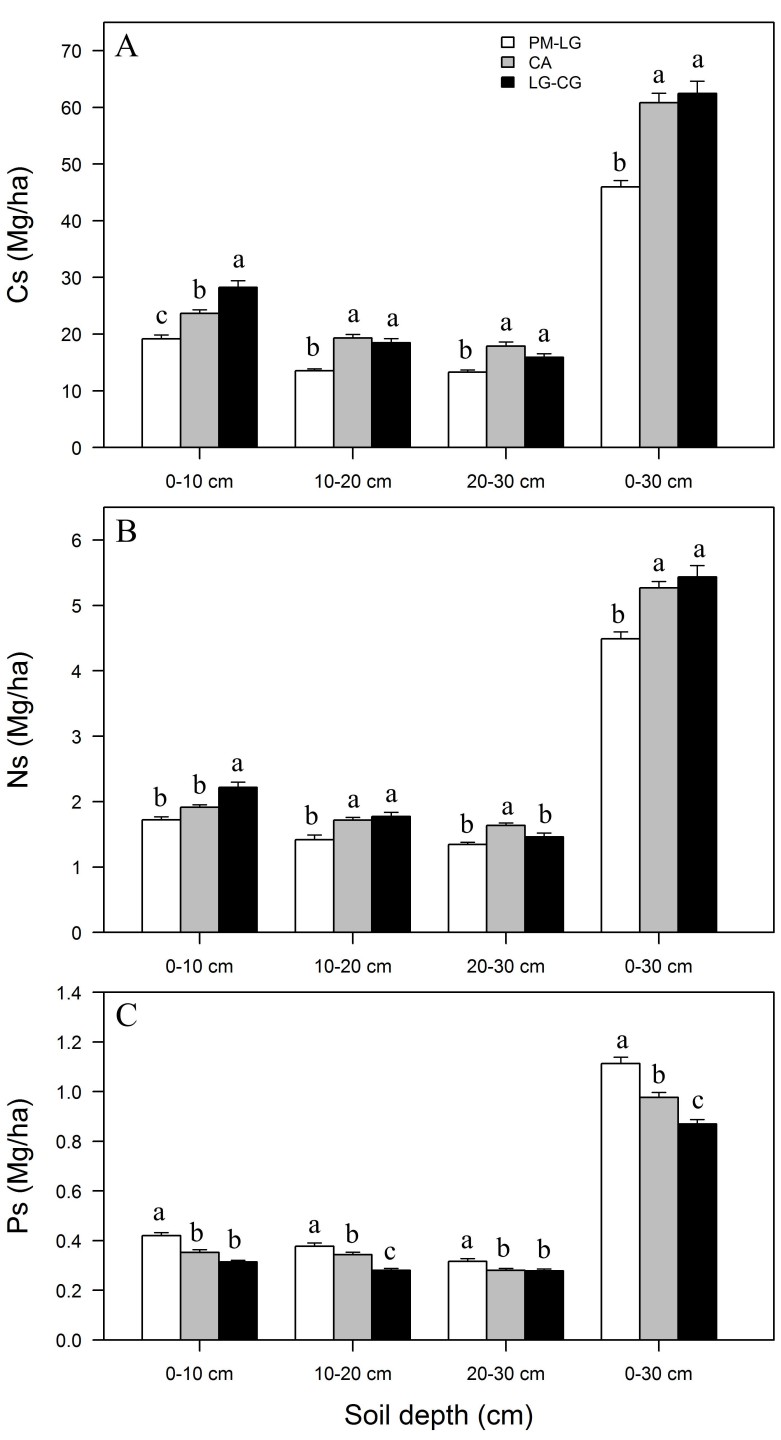