# Peer review of "Variations in soil carbon, nitrogen, phosphorus and stoichiometry along forest succession in southern China"

_Biogeosciences, 2017_

## Referee Comment (RC1) · Anonymous Referee #1 · 21 Nov 2017

Review of Ouyang, S. et al.:
Variations in soil carbon, nitrogen, phosphorus and stoichiometry along forest succession in southern China

**General comments (overall quality)**
The authors present an inventory of the soil C, N and P concentrations, storage and the soil C:N:P stoichiometry of three different forest types in southern China. The  main study question is whether/how forest succession, as represented by the three forest types, influences soil C, N and P in different soil depths. Although offering a good overview of soil C, N and P in the studied forests, the manuscript suffers from a lack of novelty and thoroughness. There are plenty of analyses of C, N and P in the mineral soil of forest ecosystems and the authors fail to point out why their study is still needed. The missing reasoning for the choice of the study topic and the lack of hypotheses (only general study aims are given) is reflected in a rather aimless discussion and the conclusions get very little support from the main body of text. The methods section does not provide all information needed to reproduce the study and the results part is sloppy. Figures and tables are frequently cited wrong. Many questions are left open, as seen in the comments below.
Before any publication, I suggest a strong revision of the manuscript because by now it is imprecise and without focus.

**Specific comments (individual scientific questions)**
*Abstract*
- Please specify the results section by presenting numbers (e.g. percentages, r values) instead of just
        stating that something significantly increased/ decreased/ correlated.
- You should end with a real conclusion. Point out why the influence of forest succession should be
        recognized/ how your results can be connected to forest ecology.

*Introduction*
Please restructure the introduction focusing on your topic "forest succession". It contains too much general knowledge and leaves too many questions about your topic. The relevance of "succession" should be clear from the first paragraph on, and the introduction should culminate in your hypotheses. I strongly suggest to dismiss the general "study aims" and implement real hypotheses because these are the core of any scientific work and likely will help to give structure to the discussion.

- Shorten and move (or delete) first paragraph (too general and no focus on study topic)
- The second paragraph seems to be a much better start for me, but it should be expanded.
        Why does forest succession improve soil C sequestration?
        How are N and P behaving controversial during forest succession?
- The third paragraph indicates a knowledge gap (soil stoichiometry of forests in southern China)
        Why does it need to be closed?
        What are the expected benefits for forest management or soil research?
        What was learned from similar analyses you cite for northern China?
- Fourth paragraph: please use hypotheses instead of "aims".
        Aim (1): What patterns do you expect?
        Aim (2): It is very difficult and speculative to assess the nutrient limitations of a forest
        without considering the vegetation (e.g. concentrations in leaves, biomass production etc.)

Aim (3): This comes as a surprise. Why do you need to look at these parameters? They were not introduced before. If they are important, you should draw the readers' attention towards them in the introduction.

*Materials and methods*
- Your choice of abbreviations for the forest types (PM-LG, CA, LG-CG) is not obvious for most readers. To use the corresponding stage of succession for each forest ("early", "middle" or "late" as in Table 2) would improve the readability of your manuscript.
- The description of the three forest types of section 2.2 should be implemented in section 2.1 because it is still site description.
- It is not clear to me how you took soil samples and determined bulk density. Please clarify the second paragraph in section 2.2. When was the sampling (year and season)?
- How was the data for stand factors (Shannon index, tree species number, ...) and topography derived? There is no information on this in your methods section.
- What are the hypotheses behind your statistical tests? How did you choose the input of your multivariate models?
- Did you only analyze the differences between succession stages per depth or also in between the depths you sampled at each stage? From your figures I guessed it was the first, but the results section left me with doubts (see below).

*Results*
You frequently refer wrongly to tables and figures, please check all the references. Table 3 is not referred to at all. Be precise and avoid vague statements (e.g. avoid stating two parameters differ significantly, but without telling how).

-3.1: You start with a very unspecific statement, referring to a table (I think you mean Table 1, not 2?) not related to the following topic. I suggest using Table 1 as additional information, not for starting paragraphs in your results (same in 3.2, 3.3)
- 3.2: This paragraph refers to Fig. 3, not Fig.2
- 3.3: Are your ratios molar or mass based? This could already be stated in the methods section and should also be included in your table/figure captions.
You refer to Fig. 2, not Fig.3 here.
Did you test for significant differences of stoichiometric ratios between the different depths of each forest? It reads like this, but there is only one sentence spent on the ANOVAs in your methods section. Did you take into account that the different depths are not independent of each other? Either you need to specify the explanation of your statistics or you cannot derive statistical comparisons between different depths - I would like more information on what you did there.
- 3.4: To me, the space taken by Table 4 and 5 is in no relation to its explanatory power. Would it be an option to give those as supplementary material? What about Table 3? Please explain the use of this table or remove it, if it is not needed.

*Discussion*
It is very hard to recognize the authors' train of thought, both in single sections and in the discussion as a whole. The focus on "forest succession" is often lost and there are many jumps between topics. Moreover, there is very little actual discussion, mostly results are just compared to the literature,

which is only part of the job. All in all, no coherent story is told here and I could not understand what the given results actually mean in the context of forest succession.

Therefore, I recommend to completely rewrite the discussion. Including hypotheses may help improving both structure and content.

Again, please check all your table and figure references, many are wrong.

- 4.1: What is STN and STP in comparison to TN and TP? These abbreviations were not introduced.

First paragraph:

I expected to read how forest type and soil depth affected SOC, TN and TP but there was no specific information given. This paragraph is very vague. It would be sound to start with a very short, pointed summary of the results this section refers to.

p. 6, l. 7ff.: Microorganisms are mainly performing litter decomposition and mineralization, and they are not mineralizing soil but organic matter. Further, how does the soil fauna come into play here? Please clarify this starting statement.

p. 8, l. 4: Did the trend decrease? Increase? Please be precise.

Second paragraph:

You state that litter and root input cause SOC and TN increases in late stages of succession, but explained only the root part. How is the litter affected?

Third paragraph:

This all comes a bit too short, you should give more details on the fate of P. The second sentence is not conclusive.

Fourth paragraph: To me, decreasing C and N concentrations with increasing soil depth occur as a well known phenomenon.

l. 19ff.: Again, did you really test depth related differences and, if so, how?

l. 24ff. In the next to last sentence you try to bridge the depth-related results and forest succession, but this does not come clear to me. How is TP in different soil depths related to succession?

l.25ff. (p.8-9): Yes, TP will be rock derived (or do you know about P fertilization in the studied forests?), but how does available P come into play here? Please clarify this statement, by now it does not explain anything.

- 4.2. :

First paragraph:

This is too much a list of literature references and too little discussion.

l. 4: Your starting sentence is again very vague. How does succession influence the ratios?

l. 5: What do "improved" ratios mean? Why do you present the highest ratios here?

Second paragraph:

What is your point here? I recognize C:N:P ratios as topic, but I do not get what you want to say about them - They are lower than elsewhere? They are stable? They determine soil processes?

l. 11: Referring to homeostatic stoichiometry of organisms and soils there are better general references. For example, you could cite Sterner & Elser, 2002 (Ecological stoichiometry: The biology of elements from molecules to the biosphere), Cleveland & Liptzin 2007 (C:N:P stoichiometry in soil: is there a "Redfield ratio" for the microbial biomass) or McGroddy et al.

2004 (Scaling of C:N:P stoichiometry in forests worldwide [...]), which is only a small choice of many more studies.

l. 12: What are the "key ecosystem characteristics"?

Third paragraph:
You switch to TP all of a sudden, leaving the discussion of the C:N ratios a stub. What is the effect of forest succession on C:N ratios? And what does that mean for the soil?
The discussion of C:P and N:P ratios is frayed and aimless. What do you want to say?

Fourth paragraph:
l. 10: I disagree with this statement. Your dataset shows quite some variation between the C:N:P ratios of forest types. Can you relate this to the results of your statistical analyses?

- 4.3:
The title of the section is imprecise, "factors" is a very vague term. Could you find a title describing the following more accurate?
How is the analysis of all these parameters connected to your topic "forest succession"?
To me it seems that the sum of parameters you include here prevented you from carefully considering and discussing each parameter on its own. Still, I am not sure whether this is a major point of this study.

*Conclusions*
- How can you differentiate the influence of forest succession and other soil/forest characteristics on the variability of SOC, TN and TP? Isn't it obvious to expect the elemental composition of soils to vary due to site-specific conditions? This conclusion seems quite superficial for me.
- The third sentence does not make sense, there is something wrong with the ratios you list. Did you mean "C:N ratios" in the second half of the sentence?
- I think it is inappropriate to speculate about the nutrient limitation of forests in a study that did not in the least consider the vegetation, e.g. as in foliar nutrient contents, growth, etc.
- Your discussion includes nothing about sustainable forest management. In what way are your results useful? What are the implications for forest management? This conclusion has no support from the manuscript.
- What about all your depth-related analyses? This major part of your analyses should be regarded in the conclusions.

*Tables/Figures*
- The caption of Figure 2 is given for Figure 3 (Molar or mass based ratios?)
- Figure 3 has no caption.
- Figures 1-3: What is the 0-30 cm depth? A mean value calculated from the other three depths? A 0-30 cm bulk sample? This needs explanation in the methods.
- Table 2: Mass based or molar ratios? Please use "C:N:P ratios" or "C:N:P  stoichiometry" instead of just "C:N:P"
- Table 3: This correlation is neither explained in the statistics section nor referred to in the results. What type of correlation analysis is this?
- Table 4: Again, what is the "0-30 cm layer"? How was this data derived?
- Table 5: Same question as to Table 4.

**Technical corrections (typos etc.)**

Using non-breaking space with units and statistical numbers will prevent awkward formatting. Here are some cases throughout the text where that happened (e.g. p. 6, l.2, l.18, p. 7, l. 2).

*Materials and Methods*

Section 2.3: Please add references for the SOC, TN and TP determination.

Section 2.4, l. 3: *i*th

Section 2.4, l. 6: replace "layer" by "depth"

Section 2.5, l. 12: introduce the abbreviation "DBH"

*Results*

Section 3.1, (p. 6), l. 5: "TP concentrations appeared to decrease." --> This is your results section - do not speculate. Please specify or skip.

Section 3.3, l. 20: please correct: [...] ratios of all three depths

Section 3.3, l. 23: please correct: (Fig. 2A)

Section 3.3 (p. 7), l. 5: replace "layer" by "depth"

Section 3.4, l. 13: replace "while" by "whereas"

*Discussion*

p.8, l. 3: (Table 1)

p8, l. 5: (Fig. 1 and Fig. 3)

p. 8, l. 6: please shorten: [...], in agreement with other studies (Your references).

p.8, l.20: Figure reference again wrong

p.8, l.24: delete "layers"

p. 8, l. 24: Figure reference wrong

p.9, l. 5: missing full stop

p.9, l.5, 6: It is more common to give only one decimal place for stoichiometric ratios.

p. 9, l. 8: C:N:P ratios

p. 9 , l. 21: You do not have a Fig. 4.

p. 10, l. 18: delete first "and", no comma before second "and"

p. 10, l. 22: "following"

p. 11, l. 19: "significantly"

Tables 4/5: pH, not PH

---

## Referee Comment (RC2) · Anonymous Referee #2 · 22 Nov 2017

This manuscript is an intensive study of soil stocks of carbon, nitrogen, and phosphorus, as well as their stoichiometric ratios, along gradients of soil depth and forest succession. The scale of the dataset is impressive, with these nutrient stocks measured in 100 subplots in each of three forest types at three separate depths. However, the paper is weakened by three factors: an absence of rigorous hypotheses/predictions, poorly described methods, and statistical analyses that come across as 'fishing expeditions' rather than hypothesis tests. Below I provide suggestions for improving each of these aspects.

1. Framing of the manuscript – the authors justify their study by discussing potential changes in nutrient stocks and C:N:P ratios along gradients of succession. However, they fail to provide any justification for *why* these parameters should change with forest succession, and they do not appear to have any directional hypotheses. I understand why this may be so: throughout the process of succession, an ecosystem will experience changes in both abiotic and biotic conditions, which could be expected to have interacting influences on belowground processes. For example, as the canopy closes, soils may experience less insolation and more buffering from temperature extremes; meanwhile, the plant community may shift in such a way that the average chemical quality of litter inputs changes. Given the complexity of these processes, is there any reason to expect that soil stocks of C, N, and P (or their ratios) should exhibit generalizable, directional shifts along successional gradients? If so, what should we expect these patterns to be? If not, then how do the results of this particular study shape our understanding of feedbacks between plant communities and soil properties during succession?

2. Experimental design: Much of the statistical analyses focus on relating C, N, and P stocks to stand-level attributes (e.g. tree community diversity) as well as topography and soil texture. Nowhere in the manuscript are these measurements described. Were they taken from another study? Were these measurements taken at the level of the 10x10 m subplot, or at the level of the three 1 ha forest plots? If the latter, the multivariate models are severely overfitted. This brings me to my third point:

3. Statistical issues: Several aspects of the statistics appear to be poorly thought out. For example, in Table 3, correlation coefficients are reported between stocks of a single nutrient (e.g. SOC) and then the ratio of SOC and TN (C:N). By definition, these variables will be highly correlated – one is derived from the other. In Tables 4 and 5, the authors report a multiple regression with no fewer than fourteen explanatory variables, several of which MUST be highly collinear (e.g. the Shannon index and species richness). This comes across as a fishing expedition, not a rigorous hypothesis test, and it is nearly impossible to interpret the results of such an analysis. Similarly, why analyze both C,N, and P concentrations AND stocks? Does the concentration data provide any insight that the stock does not?

The discussion is extensive, and makes a great deal of generalizations that are probably unwarranted (e.g. 'a low C:N ratio implies that soil organic matter is accumulating slower than it is decomposing;' 'a C:N ratio lower than 10 indicates that less organic matter is being merged into the soil.' These simplistic statements belie an understanding of how plant litter C is incorporated into SOM). Individual significant correlations are discussed, but there is no

synthesis that relates these patterns back to the specific successional trajectory of this forest ecosystem.

There is a large amount of data here, and there is absolutely the potential to say something valuable about soil nutrient cycling in relation to succession. However, the manuscript must be thoroughly revised in order to do the dataset justice.

---

## Author Comment (AC1) · 12 Dec 2017

We are grateful to the anonymous referee for the constructive comments and helpful suggestions. Based on the comments, we will revise the manuscript. Our point-by-point responses to the comments are presented below.

Q:This manuscript is an intensive study of soil stocks of carbon, nitrogen, and phosphorus, as well as their stoichiometric ratios, along gradients of soil depth and forest succession. The scale of the dataset is impressive, with these nutrient stocks measured in 100 subplots in each of three forest types at three separate depths. However, the paper is weakened by three factors: an absence of rigorous hypotheses/predictions, poorly

described methods, and statistical analyses that come across as 'fishing expeditions' rather than hypothesis tests. Below I provide suggestions for improving each of these aspects.

Re. Thanks for the positive comments and valuable suggestions on our manuscript. Based on the comments, we will propose three new hypotheses, provide clear description of methods, and use the variance inflation factor (VIF) method to remove strongly multicollinear variables (see the responses below).

Q:1. Framing of the manuscript – the authors justify their study by discussing potential changes in nutrient stocks and C:N:P ratios along gradients of succession. However, they fail to provide any justification for why these parameters should change with forest succession, and they do not appear to have any directional hypotheses. I understand why this may be so: throughout the process of succession, an ecosystem will experience changes in both abiotic and biotic conditions, which could be expected to have interacting influences on belowground processes. For example, as the canopy closes, soils may experience less insolation and more buffering from temperature extremes; meanwhile, the plant community may shift in such a way that the average chemical quality of litter inputs changes. Given the complexity of these processes, is there any reason to expect that soil stocks of C, N, and P (or their ratios) should exhibit generalizable, directional shifts along successional gradients? If so, what should we expect these patterns to be? If not, then how do the results of this particular study shape our understanding of feedbacks between plant communities and soil properties during succession?

Re. Good comments! In introduction section, we will review how and why soil stocks of C, N, and P (or their ratios) change with forest succession in the published literatures and describe the gap of this change patterns in subtropical areas of China. We will propose three hypotheses: (1) whether concentrations and storage of soil organic carbon (SOC) and total nitrogen (TN) increase but phosphorus (TP) decrease as forest succession; (2) how forest succession affects stoichiometry of soil C, N, and P and alters

nutrient limitation; and (3) what main factors influence SOC, TN, and TP concentrations and storage.

Q:2. Experimental design: Much of the statistical analyses focus on relating C, N, and P stocks to stand-level attributes (e.g. tree community diversity) as well as topography and soil texture. Nowhere in the manuscript are these measurements described. Were they taken from another study? Were these measurements taken at the level of the 10x10 m subplot, or at the level of the three 1 ha forest plots? If the latter, the multivariate models are severely overfitted. This brings me to my third point:

Re. Sorry for our unclear description. All the measurements (including stand characteristics, topography and soil properties) were taken at the level of $10\times10$ m subplot within the three forests. We will add the detailed description of the measurements.

Q:3. Statistical issues: Several aspects of the statistics appear to be poorly thought out. For example, in Table 3, correlation coefficients are reported between stocks of a single nutrient (e.g. SOC) and then the ratio of SOC and TN (C:N). By definition, these variables will be highly correlated–one is derived from the other. In Tables 4 and 5, the authors report a multiple regression with no fewer than fourteen explanatory variables, several of which MUST be highly collinear (e.g. the Shannon index and species richness). This comes across as a fishing expedition, not a rigorous hypothesis test, and it is nearly impossible to interpret the results of such an analysis. Similarly, why analyze both C, N, and P concentrations AND stocks? Does the concentration data provide any insight that the stock does not?

Re. Good suggestions! We will not use the Pearson correlation method to analyze the data and delete the Table 3. In order to figure out the factors that significantly affects the change of SOC, TN, and TP with forest succession, we will use the variance inflation factor (VIF) method to remove strongly multicollinear variables before using multiple regression models. Because nutrient concentrations and storage exhibited similar change pattern, we will select the concentrations of SOC, TN, and TP to analyze

the affecting factors. We will use one table to present the statistical analysis results, rather than two tables. Table 5 will be deleted.

Q:The discussion is extensive, and makes a great deal of generalizations that are probably unwarranted (e.g. 'a low C:N ratio implies that soil organic matter is accumulating slower than it is decomposing;' 'a C:N ratio lower than 10 indicates that less organic matter is being merged into the soil.' These simplistic statements belie an understanding of how plant litter C is incorporated into SOM). Individual significant correlations are discussed, but there is no synthesis that relates these patterns back to the specific successional trajectory of this forest ecosystem.

Re. We will rewrite the discussion section (see our response to the comments from Reviewer 1). We will delete the unwarranted sentences and Pearson correlation analysis. Based on the new hypotheses, we will focus on the successional trajectory to discuss the change patterns of SOC, TN, and TP as well as the influencing factors.

Q:There is a large amount of data here, and there is absolutely the potential to say something valuable about soil nutrient cycling in relation to succession. However, the manuscript must be thoroughly revised in order to do the dataset justice.

Re. Thanks for the positive comments. We will revise the manuscript thoroughly and hope the revision is acceptable.

―――――――――――――――――――――

---

## Author Comment (AC2) · 12 Dec 2017

We thank to the anonymous referee for the careful reading, valuable comments and constructive suggestions. We will take the suggestions into consideration when we revise the manuscript. Our detailed responses to the comments are presented below.

General comments

Q: The authors present an inventory of the soil C, N and P concentrations, storage and the soil C:N:P stoichiometry of three different forest types in southern China. The main study question is whether/how forest succession, as represented by the three forest

types, influences soil C, N and P in different soil depths. Although offering a good overview of soil C, N and P in the studied forests, the manuscript suffers from a lack of novelty and thoroughness. There are plenty of analyses of C, N and P in the mineral soil of forest ecosystems and the authors fail to point out why their study is still needed. The missing reasoning for the choice of the study topic and the lack of hypotheses (only general study aims are given) is reflected in a rather aimless discussion and the conclusions get very little support from the main body of text. The methods section does not provide all information needed to reproduce the study and the results part is sloppy. Figures and tables are frequently cited wrong. Many questions are left open, as seen in the comments below. Before any publication, I suggest a strong revision of the manuscript because by now it is imprecise and without focus.

Re. Thanks for the positive comments and critical suggestions on our manuscript. These suggestions are helpful to improve the quality of our manuscript. We will take the comments seriously when we revise the manuscript.

At first, we will clarify the objectives to test three hypotheses: (1) whether concentrations and storage of soil organic carbon (SOC) and total nitrogen (TN) increase but that of phosphorus (TP) decrease as forest succession; (2) how forest succession affects stoichiometry of soil C, N, and P and alters nutrient limitation; and (3) what main factors influence SOC, TN, and TP concentrations and storage. Even though soil nutrient content and stoichiometry changes with forest succession have been investigated in the northern Loess Plateau (Jia et al., 2012; Deng et al., 2013; 2014) and southwestern karst area (Liu et al., 2015; Zhang et al., 2015) of China, these topics have not been well understood in subtropical China. Because subtropical forests compose of diverse trees species and restore fast in southern China, the influence of forest succession on soil SOC, TN, and TP is quite different from other areas. Therefore, we hope the objectives are clearer than it was before and the hypotheses are novel.

Secondly, we will focus on the above hypotheses to revise thoroughly the entire manuscript, including the discussions, conclusions, methods, and results sections.

Thirdly, we will correct the wrong citations of the figures and tables in the text.

We hope the revision will be satisfactory for publication.

Specific comments

Abstract

Q: - Please specify the results section by presenting numbers (e.g. percentages, r values) instead of just stating that something significantly increased/ decreased/ correlated.

Re. We will add the numbers including percentages and p values as suggested.

Q: - You should end with a real conclusion. Point out why the influence of forest succession should be recognized/ how your results can be connected to forest ecology.

Re. Good suggestion! We will add the sentences "The results indicated that subtropical forest succession increased SOC and TN concentrations and storage, but decreased TP. The increases in SOC and TN concentrations is attributed to the increasing litter input due to floristic composition changes, while the decrease in TP concentrations could be explained by accumulated P in the biomass and higher P resorption as forest succession".

Introduction

Q: Please restructure the introduction focusing on your topic "forest succession". It contains too much general knowledge and leaves too many questions about your topic. The relevance of "succession" should be clear from the first paragraph on, and the introduction should culminate in your hypotheses. I strongly suggest to dismiss the general "study aims" and implement real hypotheses because these are the core of any scientific work and likely will help to give structure to the discussion.

Re. Thanks for valuable suggestions! We will revise the introduction section with emphasis on forest succession as suggested, and delete the sentences of general

knowledge. We will start to describe the effects of forest succession on soil in the first paragraph, then review how and why forest succession affect soil SOC, TN, and TP, and propose hypotheses based on the gap in the published literatures. At the same time, we will delete the general study aims.

Q: - Shorten and move (or delete) first paragraph (too general and no focus on study topic).

Re. We will refine the first paragraph.

Q: - The second paragraph seems to be a much better start for me, but it should be expanded. Why does forest succession improve soil C sequestration? How are N and P behaving controversial during forest succession?

Re. We will give more description of Why does forest succession improve soil C sequestration?" (Changes in tree species composition with forest succession result in the different amounts and quality of litter (leaf and root), and their decomposition rate, and consequently affect the soil C) and "How are N and P behaving controversial during forest succession?" (N cycling is more open than P cycling).

Q: - The third paragraph indicates a knowledge gap (soil stoichiometry of forests in southern China). Why does it need to be closed? What are the expected benefits for forest management or soil research? What was learned from similar analyses you cite for northern China?

Re. Based on the comments, we will revise the paragraph by describing difference in floristic composition, litter input and nutrient resorption in subtropical forest, compared to northern China. This work will provide the knowledge of forest management to improve the soil C sequestration and nutrients use efficiency in subtropical forests.

Q: - Fourth paragraph: please use hypotheses instead of "aims". Aim (1): What patterns do you expect? Aim (2): It is very difficult and speculative to assess the nutrient limitations of a forest without considering the vegetation (e.g. concentrations in leaves,

biomass production etc.) to reveal soil C, N, and P stoichiometry characteristics and nutrient limitation in secondary forests. Aim (3): This comes as a surprise. Why do you need to look at these parameters? They were not introduced before. If they are important, you should draw the readers' attention towards them in the introduction.

Re. We will replace the aims with hypotheses (see above). For the hypothesis 2, we will add the nutrient stoichiometry in leaves and the sentences to describe influencing factors (parameters) in the introduction section.

Materials and methods

Q: - Your choice of abbreviations for the forest types (PM-LG, CA, LG-CG) is not obvious for most readers. To use the corresponding stage of succession for each forest ("early", "middle" or "late" as in Table 2) would improve the readability of your manuscript.

Re. We will change "PM-LG, CA, LG-CG" into "early, middle, late" as suggested.

Q: - The description of the three forest types of section 2.2 should be implemented in section 2.1 because it is still site description.

Re. We will move the description of the three forest types of section 2.2 into section 2.1.

Q: - It is not clear to me how you took soil samples and determined bulk density. Please clarify the second paragraph in section 2.2. When was the sampling (year and season)?

Re. We will add the sentences to describe the methods and time (between May 25 and June 16 in 2014) to sample soil samples (within each sampling subplot, floor litter in $50 \times 50$ cm areas was collected prior to soil sampling. Subsequently, soil samples were collected at depths of 0–10, 10–20, and 20–30 cm and taken to the laboratory. After they were air dried, the soil samples were ground and passed through a 2-mm mesh sieve for physico-chemical analysis.) and measure bulk density (undisturbed

soil cores were sampled at the middle of each layer using a 5-cm diameter and 5-cm high-stainless steel cutting ring for measuring bulk density (BD). Three replicates were surveyed in each subplot. The volume of each soil corer and soil dry mass, after oven-drying at 105°C for 48 h, were measured.).

Q: - How was the data for stand factors (Shannon index, tree species number, ...) and topography derived? There is no information on this in your methods section.

Re. We will add the information about the calculation of the Shannon index, tree species number and topography in the method section. In fact, we determined tree species, diameter at breast height (DBH), height and crown width for each individual stem within 10×10 m subplots in three forests and the elevations of four corners for each subplot. These data were used to calculate the Shannon index, tree species number and topography.

Q: - What are the hypotheses behind your statistical tests? How did you choose the input of your multivariate models?

Re. We will propose hypothesis as mentioned above. At first, we will select the following parameters: stand characteristics [Shannon index, richness, density, average diameter at breast height (DBH), deciduous proportion (D), evergreen proportion (E) and litter biomass], soil bulk density (BD), soil moisture concentrations (MC), clay content, and silt content, topography variables (elevation and convexity). Second, the variance inflation factor (VIF) method was used to remove strongly multicollinear variables. At last, we will use multivariate regression model to determine what are the main factors affecting the soil SOC and nutrient.

Q: - Did you only analyze the differences between succession stages per depth or also in between the depths you sampled at each stage? From your figures I guessed it was the first, but the results section left me with doubts (see below).

Re. We used succession stages and soil depths as the explanatory variables, soil

[Figure]

nutrients as dependent variable. Two-way analysis of variance (Two-way ANOVA) was used to test the significant difference among successional stages and soil depths. We will revise the sentences in the method section.

Results

Q: You frequently refer wrongly to tables and figures, please check all the references. Table 3 is not referred to at all. Be precise and avoid vague statements (e.g. avoid stating two parameters differ significantly, but without telling how).

Re. Sorry for that figures and tables were wrongly referred. We will correct them and revise the vague sentences.

Q: -3.1: You start with a very unspecific statement, referring to a table (I think you mean Table 1, not 2?) not related to the following topic. I suggest using Table 1 as additional information, not for starting paragraphs in your results (same in 3.2, 3.3)

Re. Yes, the sentences should be referred to Table 1. So we will correct it.

Q: - 3.2: This paragraph refers to Fig. 3, not Fig.2

Re. Thanks for your careful reading. We will correct them by exchanging the captions of Fig. 3 and Fig. 2.

Q: - 3.3: Are your ratios molar or mass based? This could already be stated in the methods section and should also be included in your table/figure captions.

Re. The ratios were mass based. We will add this information to the methods section and table/figure captions.

Q: You refer to Fig. 2, not Fig.3 here.

Re. We will change "Fig. 3" into Fig. 2".

Q: Did you test for significant differences of stoichiometric ratios between the different depths of each forest? It reads like this, but there is only one sentence spent on the

ANOVAs in your methods section. Did you take into account that the different depths are not independent of each other? Either you need to specify the explanation of your statistics or you cannot derive statistical comparisons between different depths - I would like more information on what you did there.

Re. We used succession stages and soil depths as the explanatory variables, stoichiometric ratios as dependent variable. Two-way analysis of variance (Two-way ANOVA) was used to test the significant difference among successional stages and soil depths. We will revise the sentences to make it clearer in the method section.

Q: - 3.4: To me, the space taken by Table 4 and 5 is in no relation to its explanatory power. Would it be an option to give those as supplementary material? What about Table 3? Please explain the use of this table or remove it, if it is not needed.

Re. We will delete the Table 3 because we do not use Pearson relation analysis in the revised manuscript. Nutrient concentrations and storage exhibited similar change pattern, hence we will use Table 4 to present new analysis results and delete Table 5.

Discussion

Q: It is very hard to recognize the authors' train of thought, both in single sections and in the discussion as a whole. The focus on "forest succession" is often lost and there are many jumps between topics. Moreover, there is very little actual discussion, mostly results are just compared to the literature, which is only part of the job. All in all, no coherent story is told here and I could not understand what the given results actually mean in the context of forest succession. Therefore, I recommend to completely rewrite the discussion. Including hypotheses may help improving both structure and content.

Re. We will revise the discussion section thoroughly. Based on the new hypotheses, we will divide discussion section into three subsections. In each subsection, we will discuss whether our results support the hypothesis, the consistence with the literatures, and the reasons behind the results.

[Figure]

Q: Again, please check all your table and figure references, many are wrong.

Re. We will check the table and figure references carefully and correct the wrong citation in the text.

Q: - 4.1: What is STN and STP in comparison to TN and TP? These abbreviations were not introduced.

Re. Sorry for our carelessness, STN was TN and STP was TP. We will change STN into TN and change STP into TP.

First paragraph: Q: I expected to read how forest type and soil depth affected SOC, TN and TP but there was no specific information given. This paragraph is very vague. It would be sound to start with a very short, pointed summary of the results this section refers to.

Re. Good comments! We will revise the first paragraph focus on discussing how forest type and soil depth affect SOC, TN and TP.

Q: p. 6, l. 7ff.: Microorganisms are mainly performing litter decomposition and mineralization, and they are not mineralizing soil but organic matter. Further, how does the soil fauna come into play here? Please clarify this starting statement.

Re. We will change "soil mineralization" into "soil organic matter mineralization". Soil fauna could break down the litter and excretes. However, we did not conduct the relevant experiment and no fauna data available at this moment to support the statement, thus the sentence will be deleted.

Q: p. 8, l. 4: Did the trend decrease? Increase? Please be precise.

Re. We will revise the sentence and specify that SOC and TN concentrations increased, but TP decreased as forest succession.

Second paragraph: Q: You state that litter and root input cause SOC and TN increases in late stages of succession, but explained only the root part. How is the litter affected?

Re. We will add the explanation of the effects of increasing litter input on SOC and TN.

Third paragraph: Q: This all comes a bit too short, you should give more details on the fate of P. The second sentence is not conclusive.

Re. We will expand the paragraph and indicate that soil P was uptaken by plant and retained in biomass, and lower return in leaf litter due to the high P resorption before leaf fall.

Q: Fourth paragraph: To me, decreasing C and N concentrations with increasing soil depth occur as a well known phenomenon. l. 19ff.: Again, did you really test depth related differences and, if so, how?

Re. We used succession stages and soil depths as the explanatory variables, soil nutrient and stoichiometric ratios as dependent variable. Two-way analysis of variance (Two-way ANOVA) was used to test the significant difference among successional stages and soil depths. We will revise the sentences to make it clearer in the method section.

Q: l. 24ff. In the next to last sentence you try to bridge the depth-related results and forest succession, but this does not come clear to me. How is TP in different soil depths related to succession?

Re. Significant differences in TP occurred among soil depths at the early stage, but gradually disappeared as forest succession to the later stage. We will add the sentences to clarify this change pattern.

Q: l.25ff. (p.8-9): Yes, TP will be rock derived (or do you know about P fertilization in the studied forests?), but how does available P come into play here? Please clarify this statement, by now it does not explain anything.

Re. No P fertilization was applied in these natural forests investigated in this study. Beside from rock derived, litter input, root exudation and microbe would increase soil bioavailable P. We will revise the paragraph accordingly.

Q: - 4.2. : First paragraph: This is too much a list of literature references and too little discussion.

Re. We will delete some literature references and give more discussion in this part.

Q: l. 4: Your starting sentence is again very vague. How does succession influence the ratios?

Re. We will revise the sentence by describing that the ratios tended to increase as forest succession.

Q: l. 5: What do "improved" ratios mean? Why do you present the highest ratios here?

Re. "improved" ratios means "increase in C:N, C:P, and N:P ratios". The highest ratios mean that the maximum value occurred at the late stage, starting to increase from early stage to late stage. We will revise the sentences.

Second paragraph: Q: What is your point here? I recognize C:N:P ratios as topic, but I do not get what you want to say about them - They are lower than elsewhere? They are stable? They determine soil processes?

Re. Here we want to compare C:N:P ratios in this study with other studies. The results showed that the C:N:P ratios in this study was far below than the values in other area of China and globe (Table 2). These results imply that the feedbacks from living organisms could modify soil nutrient contents and result in "Redfield-like" correlations between the elemental ratio of the biota and soil in terrestrial ecosystems (Sterner and Elser 2002; Cleveland and Liptzin 2007). Soil C:N:P ratios together with vegetation stoichometry are the good indicators of soil nutrient status (Mooshammer et al., 2014; Zechmeister–Boltenstern et al., 2015). We will revise this paragraph to make it clear.

Q: l. 11: Referring to homeostatic stoichiometry of organisms and soils there are better general references. For example, you could cite Sterner & Elser, 2002 (Ecological stoichiometry: The biology of elements from molecules to the biosphere), Cleveland & Liptzin 2007 (C:N:P stoichiometry in soil: is there a "Redfield ratio" for the microbial

biomass) or McGroddy et al. 2004 (Scaling of C:N:P stoichiometry in forests worldwide [...]), which is only a small choice of many more studies.

Re. Good suggestion! We will add more citation including the references as suggested.

Q: l. 12: What are the "key ecosystem characteristics"?

Re. Sorry for our ambiguous expression, it means the nutrient cycling in forest ecosystems.

Third paragraph: Q: You switch to TP all of a sudden, leaving the discussion of the C:N ratios a stub. What is the effect of forest succession on C:N ratios? And what does that mean for the soil? The discussion of C:P and N:P ratios is frayed and aimless. What do you want to say?

Re. Based on the comments, we will revise the paragraph and discuss how forest succession affect C:N, C:P, and N:P ratios, respectively.

Fourth paragraph: Q: l. 10: I disagree with this statement. Your dataset shows quite some variation between the C:N:P ratios of forest types. Can you relate this to the results of your statistical analyses?

Re. We will not use Pearson relation analysis and Table 3 will be removed. So we will delete this paragraph. Change in C:N:P ratios with forest succession and its implication will be discussed in the second paragraph of this subsection.

Q: - 4.3: The title of the section is imprecise, "factors" is a very vague term. Could you find a title describing the following more accurate?

Re. We will change "factors" into "stand characteristics, soil pH value and physical properties".

Q: How is the analysis of all these parameters connected to your topic "forest succession"? To me it seems that the sum of parameters you include here prevented you from carefully considering and discussing each parameter on its own. Still, I am not sure

whether this is a major point of this study.

Re. We will use tree species composition (tree functional group proportion), diversity, and basal area to reflect forest succession. We will re-analyze the data and discuss the effects of these parameters on the SOC, TN, and TP contents.

Conclusions

Q: - How can you differentiate the influence of forest succession and other soil/forest characteristics on the variability of SOC, TN and TP? Isn't it obvious to expect the elemental composition of soils to vary due to site-specific conditions? This conclusion seems quite superficial for me.

Re. Forest succession change stand characteristics, leading to the change of nutrient input to soil through litter (leaves and fine roots) and nutrient uptake from soil. Hence, soil nutrients change with forest succession. We choose stand characteristics, soil pH value and physical properties as the parameters to examine the effects of forest succession on SOC, TN and TP. We will revise the conclusions section with the emphasis on changes in SOC, TN and TP with forest succession and what are the main factors affect the SOC and soil nutrient.

Q: - The third sentence does not make sense, there is something wrong with the ratios you list. Did you mean "C:N ratios" in the second half of the sentence?

Re. Yes, you are right, we will change "N:P ratios" into "C:N" ratios.

Q: - I think it is inappropriate to speculate about the nutrient limitation of forests in a study that did not in the least consider the vegetation, e.g. as in foliar nutrient contents, growth, etc.

Re. Changes in vegetation stoichiometry (i.e. foliar nutrient ratios) as forest succession will be added in discussion section (see our response above).

Q: - Your discussion includes nothing about sustainable forest management. In what

way are your results useful? What are the implications for forest management? This conclusion has no support from the manuscript.

Re. The sentence will be deleted.

Q: - What about all your depth-related analyses? This major part of your analyses should be regarded in the conclusions.

Re. We will add the sentence "Significant differences in TP occurred among soil depths at the early stage, but gradually disappeared as forest succession to the later stage.".

Tables/Figures

Q: - The caption of Figure 2 is given for Figure 3 (Molar or mass based ratios?)

Re. Sorry for our carelessness! The caption of Figure 2 will be corrected by exchanging that of Figure 2 with Figure 3. All ratios shown in this study were calculated on a mass basis.

Q: - Figure 3 has no caption.

Re. In fact, the title of Figure 2 was for Figure 3 and we will change the caption.

Q: - Figures 1-3: What is the 0-30 cm depth? A mean value calculated from the other three depths? A 0-30 cm bulk sample? This needs explanation in the methods.

Re. In the study, the SOC, TN, TP and the bulk density in the 0-30 cm soil depth were the mean of the three soil depths (0-10, 10-20, and 20-30 cm), and 0-30 cm soil stock was the sum soil stock of the three soil depths (0-10, 10-20, and 20-30 cm). We will add sentences to explain these in the methods.

Q: - Table 2: Mass based or molar ratios? Please use "C:N:P ratios" or "C:N:P stoichiometry" instead of just "C:N:P"

Re. The ratios were mass based. We will replace "C:N:P ratio" for "C:N:P".

Q: - Table 3: This correlation is neither explained in the statistics section nor referred

to in the results. What type of correlation analysis is this?

Re. Table 3 will be deleted.

Q: - Table 4: Again, what is the "0-30 cm layer"? How was this data derived?

Re. SOC, TN, and TP concentrations in 0-30 cm depth were the mean of the three soil depths (0-10, 10-20, and 20-30 cm).

Q: - Table 5: Same question as to Table 4.

Re. Table 5 will be deleted.

Technical corrections (typos etc.)

Q: Using non-breaking space with units and statistical numbers will prevent awkward formatting. Here are some cases throughout the text where that happened (e.g. p. 6, l.2, l.18, p. 7, l. 2).

Re. We will carefully check the entire manuscript and remove the space.

Materials and Methods Q: Section 2.3: Please add references for the SOC, TN and TP determination.

Re. We will add references.

Q: Section 2.4, l. 3: ith

Re. Will be changed.

Q: Section 2.4, l. 6: replace "layer" by "depth"

Re. Will be replaced.

Q: Section 2.5, l. 12: introduce the abbreviation "DBH"

Re. Will be done when DBH appears at first time (i.e. diameter at breast height).

Results

Q: Section 3.1, (p. 6), l. 5: "TP concentrations appeared to decrease." –>This is your results section - do not speculate. Please specify or skip.

Re. We will change "TP concentrations appeared to decrease" into "TP concentrations decreased".

Q: Section 3.3, l. 20: please correct: [...] ratios of all three depths

Re. Will be changed.

Q: Section 3.3, l. 23: please correct: (Fig. 2A)

Re. Will be changed.

Q: Section 3.3 (p. 7), l. 5: replace "layer" by "depth"

Re. Will be done.

Q: Section 3.4, l. 13: replace "while" by "whereas"

Re. Will be replaced.

Discussion

Q: p.8, l. 3: (Table 1)

Re. Will be corrected.

Q: p8, l. 5: (Fig. 1 and Fig. 3)

Re. Will be corrected.

Q: p. 8, l. 6: please shorten: [...], in agreement with other studies (Your references).

Re. Will be done and the references will be added.

Q: p.8, l.20: Figure reference again wrong

Re. Will be corrected.

Q: p.8, l.24: delete "layers"

Re. Will be deleted.

Q: p. 8, l. 24: Figure reference wrong

Re. Will be corrected.

Q: p.9, l. 5: missing full stop.

Re. Will be added.

Q: p.9, l.5, 6: It is more common to give only one decimal place for stoichiometric ratios.

Re. We will remain one decimal for the ratios.

Q: p. 9, l. 8: C:N:P ratios

Re. Will be changed.

Q: p. 9, l. 21: You do not have a Fig. 4.

Re. We will correct the figure reference.

Q: p. 10, l. 18: delete first "and", no comma before second "and"

Re. We will delete first "and", and add comma before second "and".

Q: p. 10, l. 22: "following"

Re. Will be changed.

Q: p. 11, l. 19: "significantly"

Re. Will be corrected.

Q: Tables 4/5: pH, not PH

Re. Will be changed.